# TopoTER: Unsupervised Learning of Topology Transformation Equivariant Representations

## Abstract

We present the Topology Transformation Equivariant Representation (TopoTER) learning, a general paradigm of unsupervised learning of node representations of graph data for the wide applicability to Graph Convolutional Neural Networks (GCNNs). We formalize the TopoTER from an information-theoretic perspective, by maximizing the mutual information between topology transformations and node representations before and after the transformations. We derive that maximizing such mutual information can be relaxed to minimizing the cross entropy between the applied topology transformation and its estimation from node representations. In particular, we seek to sample a subset of node pairs from the original graph and flip the edge connectivity between each pair to transform the graph topology. Then, we self-train a representation encoder to learn node representations by reconstructing the topology transformations from the feature representations of the original and transformed graphs. In experiments, we apply the TopoTER to the downstream node and graph classification tasks, and results show that the TopoTER outperforms the state-of-the-art unsupervised approaches.

## 1 Introduction

Graphs provide a natural and efficient representation for non-Euclidean data, such as brain networks, social networks, citation networks, and 3D point clouds. Graph Convolutional Neural Networks (GCNNs) (Bronstein et al., 2017) have been proposed to generalize the CNNs to learn representations from non-Euclidean data, which has made significant advances in various applications such as node classification (Kipf & Welling, 2017; Veličković et al., 2018; Xu et al., 2019a) and graph classification (Xu et al., 2019b). However, most existing GCNNs are trained in a supervised fashion, requiring a large amount of labeled data for network training. This limits the applications of the GCNNs since it is often costly to collect adequately labeled data, especially on large-scale graphs. Hence, this motivates the proposed research to learn graph feature representations in an unsupervised fashion, which enables the discovery of intrinsic graph structures and thus adapts to various downstream tasks.

Auto-Encoders (AEs) and Generative Adversarial Networks (GANs) are two most representative unsupervised learning methods. Based on the AEs and GANs, many approaches have sought to learn *transformation equivariant representations (TERs)* to further improve the quality of unsupervised representation learning. It assumes that the learned representations equivarying to transformations are able to encode the intrinsic structures of data such that the transformations can be reconstructed from the representations before and after transformations (Qi et al., 2019b). Learning TERs traces back to Hinton's seminal work on learning transformation capsules (Hinton et al., 2011), and embodies a variety of methods developed for Euclidean data (Kivinen & Williams, 2011; Sohn & Lee, 2012; Schmidt & Roth, 2012; Skibbe, 2013; Lenc & Vedaldi, 2015; Gens & Domingos, 2014; Dieleman et al., 2015; 2016; Zhang et al., 2019; Qi et al., 2019a). Further, Gao et al. (2020) extend transformation equivariant representation learning to non-Euclidean domain, which formalizes Graph Transformation Equivariant Representation (GraphTER) learning by auto-encoding node-wise transformations in an unsupervised fashion. Nevertheless, only transformations on node features are explored, while the underlying graph may vary implicitly. The graph topology has not been fully explored yet, which however is crucial in unsupervised graph representation learning.

To this end, we propose the Topology Transformation Equivariant Representation (TopoTER) learning to infer unsupervised graph feature representations by estimating topology transformations. In-

stead of transforming node features as in the GraphTER, the proposed TopoTER studies the transformation equivariant representation learning by transforming the graph topology, *i.e.*, adding or removing edges to perturb the graph structure. Then the same input signals are attached to the resultant graph topologies, resulting in different graph representations. This provides an insight into how the same input signals associated with different graph topologies would lead to equivariant representations enabling the fusion of node feature and graph topology in GCNNs. Formally, we propose the TopoTER from an information-theoretic perspective, aiming to maximize the mutual information between topology transformations and feature representations with respect to the original and transformed graphs. We derive that maximizing such mutual information can be relaxed to the cross entropy minimization between the applied topology transformations and the estimation from the learned representations of graph data under the topological transformations.

Specifically, given an input graph and its associated node features, we first sample a subset of node pairs from the graph and flip the edge connectivity between each pair at a perturbation rate, leading to a transformed graph with attached node features. Then, we design a graph-convolutional auto-encoder architecture, where the encoder learns the node-wise representations over the original and transformed graphs respectively, and the decoder predicts the topology transformations of edge connectivity from both representations by minimizing the cross entropy between the applied and estimated transformations. Experimental results demonstrate that the proposed TopoTER model outperforms the state-of-the-art unsupervised models, and even achieves comparable results to the (semi-)supervised approaches in node classification and graph classification tasks at times.

Our main contributions are summarized as follows.

- We propose the Topology Transformation Equivariant Representation (TopoTER) learning to infer expressive node feature representations in an unsupervised fashion, which can characterize the intrinsic structures of graphs and the associated features by exploring the graph transformations of connectivity topology.

- We formulate the TopoTER from an information-theoretic perspective, by maximizing the mutual information between feature representations and topology transformations, which can be relaxed to the cross entropy minimization between the applied transformations and the prediction in an end-to-end graph-convolutional auto-encoder architecture.

- Experiments demonstrate that the proposed TopoTER model outperforms the state-of-the-art unsupervised methods in both node classification and graph classification.

## 2 RELATED WORK

**Graph Auto-Encoders.** Graph Auto-Encoders (GAEs) are the most representative unsupervised methods. GAEs encode graph data into feature space via an encoder and reconstruct the input graph data from the encoded feature representations via a decoder. GAEs are often used to learn network embeddings and graph generative distributions (Wu et al., 2020). For network embedding learning, GAEs learn the feature representations of each node by reconstructing graph structural information, such as the graph adjacency matrix (Kipf & Welling, 2016) and the positive pointwise mutual information (PPMI) matrix (Cao et al., 2016; Wang et al., 2016). For graph generation, some methods generate nodes and edges of a graph alternately (You et al., 2018), while other methods output an entire graph (Simonovsky & Komodakis, 2018; Ma et al., 2018; De Cao & Kipf, 2018).

**Graph Contrastive Learning.** An important paradigm called contrastive learning aims to train an encoder to be *contrastive* between the representations of positive samples and negative samples. Recent contrastive learning frameworks can be divided into two categories (Liu et al., 2020): *context-instance* contrast and *context-context* contrast. Context-instance contrast focuses on modeling the relationships between the local feature of a sample and its global context representation. Deep InfoMax (DIM) (Hjelm et al., 2018) first proposes to maximize the mutual information between a local patch and its global context through a contrastive learning task. Deep Graph InfoMax (DGI) (Velickovic et al., 2019) proposes to learn node-level feature representation by extending DIM to graph-structured data, while InfoGraph (Sun et al., 2020a) aims to use mutual information maximization for unsupervised representation learning on entire graphs. Peng et al. (2020) propose a Graphical Mutual Information (GMI) approach to maximize the mutual information of both features and edges between inputs and outputs. In contrast to context-instance methods, context-context contrast studies the relationships between the global representations of different samples. M3S (Sun et al., 2020b) adopts a self-supervised pre-training paradigm as in DeepCluster (Caron et al., 2018) for better semi-supervised prediction in GCNNs. Graph Contrastive Coding (GCC)

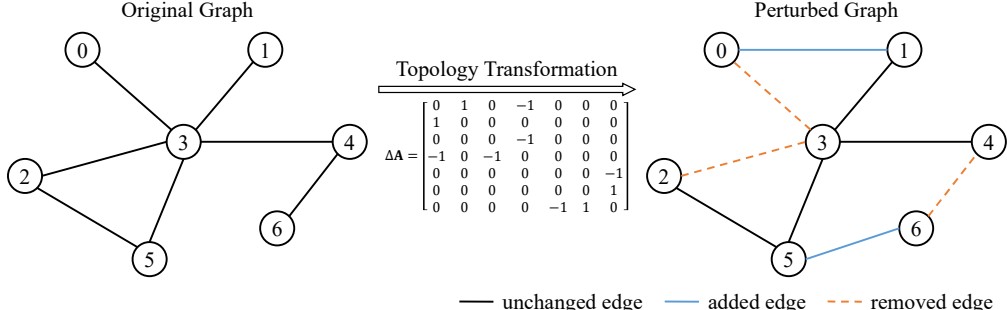

Figure 1: An example of graphs before and after topology transformations.

(Qiu et al., 2020) designs the pre-training task as subgraph instance discrimination in and across networks to empower graph neural networks to learn the intrinsic structural representations.

**Transformation Equivariant Representation Learning.** Many approaches have sought to learn transformation equivariant representations. Learning transformation equivariant representations has been advocated in Hinton's seminal work on learning transformation capsules. Following this, a variety of approaches have been proposed to learn transformation equivariant representations (Gens & Domingos, 2014; Dieleman et al., 2015; 2016; Cohen & Welling, 2016; Lenssen et al., 2018). To generalize to generic transformations, Zhang et al. (2019) propose to learn unsupervised feature representations via Auto-Encoding Transformations (AET) by estimating transformations from the learned feature representations of both the original and transformed images, while Qi et al. (2019a) extend AET from an information-theoretic perspective by maximizing the lower bound of mutual information between transformations and representations. Wang et al. (2020) extend the AET to Generative Adversarial Networks (GANs) for unsupervised image synthesis and representation learning. Gao et al. (2020) introduce the GraphTER model that extends AET to graph-structured data, which is formalized by auto-encoding node-wise transformations in an unsupervised manner. de Haan et al. (2020) propose Gauge Equivariant Mesh CNNs which generalize GCNNs to apply anisotropic gauge equivariant kernels. Fuchs et al. (2020) introduce a self-attention mechanism specifically for 3D point cloud data, which adheres to equivariance constraints, improving robustness to nuisance transformations.

## 3 METHOD

### 3.1 PRELIMINARY

We consider an undirected graph $\mathcal{G} = \{\mathcal{V}, \mathcal{E}, \mathbf{A}\}$ composed of a node set $\mathcal{V}$ of cardinality $|\mathcal{V}| = N$, an edge set $\mathcal{E}$ connecting nodes of cardinality $|\mathcal{E}| = M$. $\mathbf{A}$ is a real symmetric $N \times N$ matrix that encodes the graph structure, where $a_{i,j} = 1$ if there exists an edge $(i, j)$ between nodes $i$ and $j$, and $a_{i,j} = 0$ otherwise. *Graph signal* refers to data that reside on the nodes of a graph $\mathcal{G}$, denoted by $\mathbf{X} \in \mathbb{R}^{N \times C}$ with the $i$-th row representing the $C$-dimensional graph signal on the $i$-th node of $\mathcal{V}$.

### 3.2 TOPOLOGY TRANSFORMATION

We define the topology transformation $\mathbf{t}$ as adding or removing edges from the original edge set $\mathcal{E}$ in graph $\mathcal{G}$. This can be done by sampling, i.i.d., a *switch* parameter $\sigma_{i,j}$ as in (Velickovic et al., 2019), which determines whether to modify edge $(i, j)$ in the adjacency matrix. Assuming a Bernoulli distribution $\mathcal{B}(p)$, where $p$ denotes the probability of each edge being modified, we draw a random matrix $\Sigma = \{\sigma_{i,j}\}_{N \times N}$ from $\mathcal{B}(p)$, *i.e.*, $\Sigma \sim \mathcal{B}(p)$. We then acquire the perturbed adjacency matrix as

$$\widetilde{\mathbf{A}} = \mathbf{A} \oplus \Sigma, \tag{1}$$

where $\oplus$ is the exclusive OR (XOR) operation. This strategy produces a transformed graph through the topology transformation $\mathbf{t}$, *i.e.*, $\widetilde{\mathbf{A}} = \mathbf{t}(\mathbf{A})$. Here, the edge perturbation probability of $p = 0$ corresponds to a non-transformed adjacency matrix, which is a special case of an identity transformation to $\mathbf{A}$.

The transformed adjacency matrix $\widetilde{\mathbf{A}}$ can also be written as the sum of the original adjacency matrix $\mathbf{A}$ and a topology perturbation matrix $\Delta\mathbf{A}$:

$$\widetilde{\mathbf{A}} = \mathbf{A} + \Delta\mathbf{A}, \tag{2}$$

where $\Delta\mathbf{A} = \{\delta a_{i,j}\}_{N \times N}$ encodes the perturbation of edges, with $\delta a_{i,j} \in \{-1, 0, 1\}$. As shown in Fig. 1, when $\delta a_{i,j} = 0$, the edge between node $i$ and node $j$ keeps unchanged (*i.e.*, black solid lines); when $\delta a_{i,j} = -1$ or $1$, it means removing (*i.e.*, orange dotted lines) or adding (*i.e.*, blue solid lines) the edge between node $i$ and node $j$, respectively.

### 3.3 THE FORMULATION OF TOPOTER

**Definition 1** *Given a pair of graph signal and adjacency matrix* $(\mathbf{X}, \mathbf{A})$*, and a pair of graph signal and transformed adjacency matrix* $(\mathbf{X}, \widetilde{\mathbf{A}})$ *by a topology transformation* $\mathbf{t}(\cdot)$*, a function* $E(\cdot)$ *is transformation equivariant if it satisfies*

$$E(\mathbf{X}, \widetilde{\mathbf{A}}) = E(\mathbf{X}, \mathbf{t}(\mathbf{A})) = \rho(\mathbf{t})\left[E(\mathbf{X}, \mathbf{A})\right], \tag{3}$$

*where* $\rho(\mathbf{t})[\cdot]$ *is a homomorphism of transformation* $\mathbf{t}$ *in the representation space.*

Let us denote $\mathbf{H} = E(\mathbf{X}, \mathbf{A})$, and $\widetilde{\mathbf{H}} = E(\mathbf{X}, \widetilde{\mathbf{A}})$. We seek to learn an encoder $E : (\mathbf{X}, \mathbf{A}) \mapsto \mathbf{H}; (\mathbf{X}, \widetilde{\mathbf{A}}) \mapsto \widetilde{\mathbf{H}}$ that maps both the original and transformed sample to representations $\{\mathbf{H}, \widetilde{\mathbf{H}}\}$ equivariant to the sampled transformation $\mathbf{t}$, whose information can thus be inferred from the representations via a decoder $D : (\widetilde{\mathbf{H}}, \mathbf{H}) \mapsto \widehat{\Delta\mathbf{A}}$ as much as possible. From an information-theoretic perspective, this requires $(\mathbf{H}, \Delta\mathbf{A})$ should jointly contain all necessary information about $\widetilde{\mathbf{H}}$.

Then a natural choice to formalize the topology transformation equivariance is the *mutual information* $I(\mathbf{H}, \Delta\mathbf{A}; \widetilde{\mathbf{H}})$ between $(\mathbf{H}, \Delta\mathbf{A})$ and $\widetilde{\mathbf{H}}$. The larger the mutual information is, the more knowledge about $\Delta\mathbf{A}$ can be inferred from the representations $\{\mathbf{H}, \widetilde{\mathbf{H}}\}$. Hence, we propose to maximize the mutual information to learn the topology transformation equivariant representations as follows:

$$\max_{\theta} \; I(\mathbf{H}, \Delta\mathbf{A}; \widetilde{\mathbf{H}}), \tag{4}$$

where $\theta$ denotes the parameters of the auto-encoder network.

Nevertheless, it is difficult to compute the mutual information directly. Instead, we derive that maximizing the mutual information can be relaxed to minimizing the cross entropy, as described in the following theorem.

**Theorem 1** *The maximization of the mutual information* $I(\mathbf{H}, \Delta\mathbf{A}; \widetilde{\mathbf{H}})$ *can be relaxed to the minimization of the cross entropy* $H(p \parallel q)$ *between the probability distributions* $p(\Delta\mathbf{A}, \widetilde{\mathbf{H}}, \mathbf{H})$ *and* $q(\widehat{\Delta\mathbf{A}}|\widetilde{\mathbf{H}}, \mathbf{H})$*:*

$$\min_{\theta} \; H\left(p(\Delta\mathbf{A}, \widetilde{\mathbf{H}}, \mathbf{H}) \parallel q(\widehat{\Delta\mathbf{A}}|\widetilde{\mathbf{H}}, \mathbf{H})\right) \triangleq - \mathop{\mathbb{E}}_{p(\Delta\mathbf{A}, \widetilde{\mathbf{H}}, \mathbf{H})} \log q(\widehat{\Delta\mathbf{A}}|\widetilde{\mathbf{H}}, \mathbf{H}). \tag{5}$$

*Proof* By using the chain rule of mutual information, we have

$$I(\mathbf{H}, \Delta\mathbf{A}; \widetilde{\mathbf{H}}) = I(\Delta\mathbf{A}; \widetilde{\mathbf{H}}|\mathbf{H}) + I(\mathbf{H}; \widetilde{\mathbf{H}}) \geq I(\Delta\mathbf{A}; \widetilde{\mathbf{H}}|\mathbf{H}).$$

Thus the mutual information $I(\Delta\mathbf{A}; \widetilde{\mathbf{H}}|\mathbf{H})$ is the lower bound of the mutual information $I(\mathbf{H}, \Delta\mathbf{A}; \widetilde{\mathbf{H}})$ that attains its minimum value when $I(\mathbf{H}; \widetilde{\mathbf{H}}) = 0$.

Therefore, we relax the objective to maximizing the lower bound mutual information $I(\Delta\mathbf{A}; \widetilde{\mathbf{H}}|\mathbf{H})$ between the transformed representation $\widetilde{\mathbf{H}}$ and the topology transformation $\Delta\mathbf{A}$:

$$I(\Delta\mathbf{A}; \widetilde{\mathbf{H}}|\mathbf{H}) = H(\Delta\mathbf{A}|\mathbf{H}) - H(\Delta\mathbf{A}|\widetilde{\mathbf{H}}, \mathbf{H}),$$

where $H(\cdot)$ denotes the conditional entropy. Since $\Delta\mathbf{A}$ and $\mathbf{H}$ are independent, we have $H(\Delta\mathbf{A}|\mathbf{H}) = H(\Delta\mathbf{A})$. Hence, maximizing $I(\Delta\mathbf{A}; \widetilde{\mathbf{H}}|\mathbf{H})$ becomes

$$\min_{\theta} \; H(\Delta\mathbf{A}|\widetilde{\mathbf{H}}, \mathbf{H}). \tag{6}$$

According to the chain rule of conditional entropy, we have

$$H(\Delta\mathbf{A}|\widetilde{\mathbf{H}}, \mathbf{H}) = H(\Delta\mathbf{A}, \widetilde{\mathbf{H}}, \mathbf{H}) - H(\widetilde{\mathbf{H}}, \mathbf{H}) \leq H(\Delta\mathbf{A}, \widetilde{\mathbf{H}}, \mathbf{H}),$$

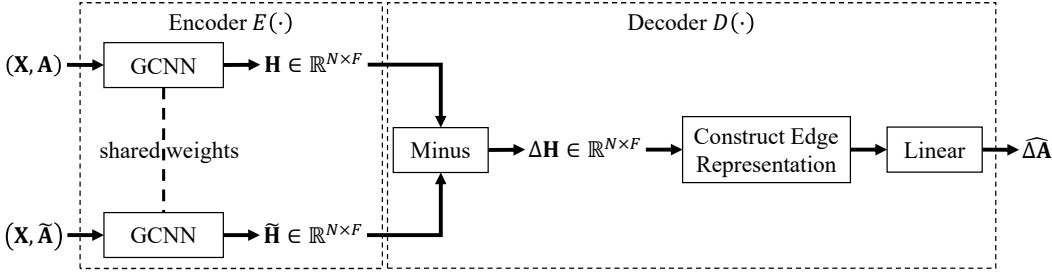

Figure 2: The architecture of the proposed TopoTER.

where the conditional entropy $H(\Delta\mathbf{A}|\widetilde{\mathbf{H}}, \mathbf{H})$ is upper bounded by the joint entropy $H(\Delta\mathbf{A}, \widetilde{\mathbf{H}}, \mathbf{H})$. Thus, the minimization problem in Eq. (6) becomes

$$\min_{\theta} \; H(\Delta\mathbf{A}, \widetilde{\mathbf{H}}, \mathbf{H}). \tag{7}$$

We next introduce a conditional probability distribution $q(\widehat{\Delta\mathbf{A}}|\widetilde{\mathbf{H}}, \mathbf{H})$ to approximate the intractable posterior $\tilde{q}(\Delta\mathbf{A}|\widetilde{\mathbf{H}}, \mathbf{H})$ with an estimated $\widehat{\Delta\mathbf{A}}$. According to the definition of the Kullback-Leibler divergence, we have

$$H(\Delta\mathbf{A}, \widetilde{\mathbf{H}}, \mathbf{H}) = H(p) = H(p \parallel q) - D_{\mathrm{KL}}(p \parallel q) \leq H(p \parallel q),$$

where $D_{\mathrm{KL}}(p \parallel q)$ denotes the Kullback-Leibler divergence of $p$ and $q$ that is non-negative, and $H(p \parallel q)$ is the cross entropy between $p$ and $q$. Thus, Eq. (6) is converted to minimizing the cross entropy as the upper bound:

$$\min_{\theta} \; H\left(p(\Delta\mathbf{A}, \widetilde{\mathbf{H}}, \mathbf{H}) \parallel q(\widehat{\Delta\mathbf{A}}|\widetilde{\mathbf{H}}, \mathbf{H})\right) \triangleq - \mathbb{E}_{p(\Delta\mathbf{A}, \widetilde{\mathbf{H}}, \mathbf{H})} \log q(\widehat{\Delta\mathbf{A}}|\widetilde{\mathbf{H}}, \mathbf{H}).$$

Hence, we relax the maximization problem in Eq. (4) to the optimization in Eq. (5). □

Based on **Theorem 1**, we train the decoder $D$ to learn the distribution $q(\widehat{\Delta\mathbf{A}}|\widetilde{\mathbf{H}}, \mathbf{H})$ so as to estimate the topology transformation $\widehat{\Delta\mathbf{A}}$ from the encoded $\{\widetilde{\mathbf{H}}, \mathbf{H}\}$, where the input pairs of original and transformed graph representations $\{\widetilde{\mathbf{H}}, \mathbf{H}\}$ as well as the ground truth target $\Delta\mathbf{A}$ can be sampled tractably from the factorization of $p(\Delta\mathbf{A}, \widetilde{\mathbf{H}}, \mathbf{H}) \triangleq p(\Delta\mathbf{A})p(\mathbf{H})p(\widetilde{\mathbf{H}}|\Delta\mathbf{A}, \mathbf{H})$. This allows us to minimize the *cross entropy* between $p(\Delta\mathbf{A}, \widetilde{\mathbf{H}}, \mathbf{H})$ and $q(\widehat{\Delta\mathbf{A}}|\widetilde{\mathbf{H}}, \mathbf{H})$ as in (5) with the training triplets $(\widetilde{\mathbf{H}}, \mathbf{H}; \Delta\mathbf{A})$ drawn from the tractable factorization of $p(\Delta\mathbf{A}, \widetilde{\mathbf{H}}, \mathbf{H})$. Hence, we formulate the TopoTER as the joint optimization of the representation encoder $E$ and the transformation decoder $D$.

## 3.4 THE ALGORITHM

We design a graph-convolutional auto-encoder network for the TopoTER learning, as illustrated in Fig. 2. Given a graph signal $\mathbf{X}$ associated with a graph $\mathcal{G} = \{\mathcal{V}, \mathcal{E}, \mathbf{A}\}$, the proposed unsupervised learning algorithm for the TopoTER consists of three steps: 1) topology transformation, which samples and perturbs some edges from $\mathcal{E}$ to acquire a transformed adjacency matrix $\widetilde{\mathbf{A}}$; 2) representation encoding, which extracts the feature representations of graph signals before and after the topology transformation; 3) transformation decoding, which estimates the topology transformation parameters from the learned feature representations. We elaborate on the three steps as follows.

**Topology Transformation.** We randomly sample a subset of edges from $\mathcal{E}$ for topology perturbation—adding or removing edges, which not only enables to characterize local graph structures at various scales, but also reduces the number of edge transformation parameters to estimate for computational efficiency. In practice, in each iteration of training, we sample *all* the node pairs with connected edges $\mathbf{S}_1$, and randomly sample a subset of disconnected node pairs $\mathbf{S}_0$, *i.e.*,

$$\mathbf{S}_0 = \left\{(i,j)\big|a_{i,j} = 0\right\}, \mathbf{S}_1 = \left\{(i,j)\big|a_{i,j} = 1\right\}, \tag{8}$$

where $|\mathbf{S}_0| = |\mathbf{S}_1| = M$. Next, we randomly split $\mathbf{S}_0$ and $\mathbf{S}_1$ into two disjoint sets, respectively, *i.e.*,

$$\mathbf{S}_i = \left\{\mathbf{S}_i^{(1)}, \mathbf{S}_i^{(2)} \; \big| \; \mathbf{S}_i^{(1)} \cap \mathbf{S}_i^{(2)} = \varnothing, \mathbf{S}_i^{(1)} \cup \mathbf{S}_i^{(2)} = \mathbf{S}_i, |\mathbf{S}_i^{(1)}| = r \cdot |\mathbf{S}_i|\right\}, i \in \{0, 1\}, \tag{9}$$

where $r$ is the *edge perturbation rate*. Then, for each node pair $(i, j)$ in $\mathbf{S}_0^{(1)}$ and $\mathbf{S}_1^{(1)}$, we *flip* the corresponding entry in the original graph adjacency matrix. That is, if $a_{i,j} = 0$, then we set $\tilde{a}_{i,j} = 1$; otherwise, we set $\tilde{a}_{i,j} = 0$. For each node pair $(i, j)$ in $\mathbf{S}_0^{(2)}$ and $\mathbf{S}_1^{(2)}$, we keep the original connectivities unchanged, *i.e.*, $\tilde{a}_{i,j} = a_{i,j}$.

This leads to the transformed adjacency matrix $\widetilde{\mathbf{A}}$, as well as the sampled transformation parameters by accessing $\Delta\mathbf{A}$ at position $(i, j)$ from $\mathbf{S}_0$ and $\mathbf{S}_1$. Also, we can category the sampled topology transformation parameters into four types:

1. add an edge to a disconnected node pair, *i.e.*, $\{\mathbf{t} : a_{i,j} = 0 \mapsto \tilde{a}_{i,j} = 1, (i, j) \in \mathbf{S}_0^{(1)}\}$;

2. delete the edge between a connected node pair, *i.e.*, $\{\mathbf{t} : a_{i,j} = 1 \mapsto \tilde{a}_{i,j} = 0, (i, j) \in \mathbf{S}_1^{(1)}\}$;

3. keep the disconnection between node pairs in $\mathbf{S}_0^{(2)}$, *i.e.*, $\{\mathbf{t} : a_{i,j} = 0 \mapsto \tilde{a}_{i,j} = 0, (i, j) \in \mathbf{S}_0^{(2)}\}$;

4. keep the connection between node pairs in $\mathbf{S}_1^{(2)}$, *i.e.*, $\{\mathbf{t} : a_{i,j} = 1 \mapsto \tilde{a}_{i,j} = 1, (i, j) \in \mathbf{S}_1^{(2)}\}$.

Thus, we cast the problem of estimating transformation parameters in $\Delta\mathbf{A}$ from $(\widetilde{\mathbf{H}}, \mathbf{H})$ as the classification problem of the transformation parameter types. The percentage of these four types is $r : r : (1 - r) : (1 - r)$.

**Representation Encoder.** We train an encoder $E : (\mathbf{X}, \mathbf{A}) \mapsto E(\mathbf{X}, \mathbf{A})$ to encode the feature representations of each node in the graph. As demonstrated in Fig. 2, we leverage GCNNs with shared weights to extract feature representations of each node in the graph signal. Taking the GCN (Kipf & Welling, 2017) as an example, the graph convolution in the GCN is defined as

$$\mathbf{H} = E(\mathbf{X}, \mathbf{A}) = \mathbf{D}^{-\frac{1}{2}}(\mathbf{A} + \mathbf{I})\mathbf{D}^{-\frac{1}{2}}\mathbf{X}\mathbf{W}, \tag{10}$$

where $\mathbf{D}$ is the degree matrix of $\mathbf{A} + \mathbf{I}$, $\mathbf{W} \in \mathbb{R}^{C \times F}$ is a learnable parameter matrix, and $\mathbf{H} = [\mathbf{h}_1, ..., \mathbf{h}_N]^\top \in \mathbb{R}^{N \times F}$ denotes the node-wise feature matrix with $F$ output channels. Similarly, the node feature of the transformed counterpart is as follows with the shared weights $\mathbf{W}$.

$$\begin{aligned}
\widetilde{\mathbf{H}} = E(\mathbf{X}, \widetilde{\mathbf{A}}) &= \widetilde{\mathbf{D}}^{-\frac{1}{2}}(\widetilde{\mathbf{A}} + \mathbf{I})\widetilde{\mathbf{D}}^{-\frac{1}{2}}\mathbf{X}\mathbf{W} \\
&= \widetilde{\mathbf{D}}^{-\frac{1}{2}}(\mathbf{A} + \mathbf{I})\widetilde{\mathbf{D}}^{-\frac{1}{2}}\mathbf{X}\mathbf{W} + \widetilde{\mathbf{D}}^{-\frac{1}{2}}\Delta\mathbf{A}\widetilde{\mathbf{D}}^{-\frac{1}{2}}\mathbf{X}\mathbf{W}.
\end{aligned} \tag{11}$$

We thus acquire the feature representations $\mathbf{H}$ and $\widetilde{\mathbf{H}}$ of graph signals before and after topology transformations.

**Transformation Decoder.** Comparing Eq. (10) and Eq. (11), the prominent difference between $\widetilde{\mathbf{H}}$ and $\mathbf{H}$ lies in the second term of Eq. (11) featuring $\Delta\mathbf{A}$. This enables us to train a decoder $D : (\widetilde{\mathbf{H}}, \mathbf{H}) \mapsto \widehat{\Delta\mathbf{A}}$ to estimate the topology transformation from the joint representations before and after transformation. We first take the difference between the extracted feature representations before and after transformations along the feature channel,

$$\Delta\mathbf{H} = \widetilde{\mathbf{H}} - \mathbf{H} = [\delta\mathbf{h}_1, ..., \delta\mathbf{h}_N]^\top \in \mathbb{R}^{N \times F}. \tag{12}$$

Thus, we can predict the topology transformation between node $i$ and node $j$ through the node-wise feature difference $\Delta\mathbf{H}$ by constructing the *edge representation* as

$$\mathbf{e}_{i,j} = \frac{\exp\{-(\delta\mathbf{h}_i - \delta\mathbf{h}_j) \odot (\delta\mathbf{h}_i - \delta\mathbf{h}_j)\}}{\|\exp\{-(\delta\mathbf{h}_i - \delta\mathbf{h}_j) \odot (\delta\mathbf{h}_i - \delta\mathbf{h}_j)\}\|_1} \in \mathbb{R}^F, \quad \forall(i, j) \in \mathbf{S}_0 \cup \mathbf{S}_1, \tag{13}$$

where $\odot$ denotes the Hadamard product of two vectors to capture the feature representation, and $\|\cdot\|_1$ is the $\ell_1$-norm of a vector for normalization. The edge representation $\mathbf{e}_{i,j}$ of node $i$ and $j$ is then fed into several linear layers for the prediction of the topology transformation,

$$\widehat{\mathbf{y}}_{i,j} = \text{softmax}\left(\text{linear}(\mathbf{e}_{i,j})\right), \quad \forall(i, j) \in \mathbf{S}_0 \cup \mathbf{S}_1, \tag{14}$$

where $\text{softmax}(\cdot)$ is an activation function.

According to Eq. (5), the entire auto-encoder network is trained by minimizing the cross entropy

$$\mathcal{L} = -\mathop{\mathbb{E}}_{(i,j)\in\mathbf{S}_0\cup\mathbf{S}_1} \sum_{f=0}^{3} \mathbf{y}_{i,j}^{(f)} \log \widehat{\mathbf{y}}_{i,j}^{(f)}, \tag{15}$$

where $f$ denotes the transformation type ($f \in \{0, 1, 2, 3\}$), and $\mathbf{y}$ is the ground-truth binary indicator (0 or 1) for each transformation parameter type.

Table 1: Node classification accuracies (with standard deviation) in percentage on three datasets. $\mathbf{X}$, $\mathbf{A}$, $\mathbf{Y}$ denote the input data, adjacency matrix and labels respectively.

| Method | Training Data | Cora | Citeseer | Pubmed |
|---|---|---|---|---|
| **Semi-Supervised Methods** | | | | |
| GCN (Kipf & Welling, 2017) | $\mathbf{X, A, Y}$ | 81.5 | 70.3 | 79.0 |
| MoNet (Monti et al., 2017) | $\mathbf{X, A, Y}$ | $81.7 \pm 0.5$ | - | $78.8 \pm 0.3$ |
| GAT (Veličković et al., 2018) | $\mathbf{X, A, Y}$ | $83.0 \pm 0.7$ | $72.5 \pm 0.7$ | $79.0 \pm 0.3$ |
| SGC (Wu et al., 2019) | $\mathbf{X, A, Y}$ | $81.0 \pm 0.0$ | $71.9 \pm 0.1$ | $78.9 \pm 0.0$ |
| GWNN (Xu et al., 2019a) | $\mathbf{X, A, Y}$ | 82.8 | 71.7 | 79.1 |
| MixHop (Abu-El-Haija et al., 2019) | $\mathbf{X, A, Y}$ | $81.9 \pm 0.4$ | $71.4 \pm 0.8$ | $80.8 \pm 0.6$ |
| DFNet (Wijesinghe & Wang, 2019) | $\mathbf{X, A, Y}$ | $85.2 \pm 0.5$ | $74.2 \pm 0.3$ | $84.3 \pm 0.4$ |
| **Unsupervised Methods** | | | | |
| Raw Features (Velickovic et al., 2019) | $\mathbf{X}$ | $47.9 \pm 0.4$ | $49.3 \pm 0.2$ | $69.1 \pm 0.3$ |
| DeepWalk (Perozzi et al., 2014) | $\mathbf{A}$ | 67.2 | 43.2 | 65.3 |
| DeepWalk + Features (Velickovic et al., 2019) | $\mathbf{X, A}$ | $70.7 \pm 0.6$ | $51.4 \pm 0.5$ | $74.3 \pm 0.9$ |
| GAE (Kipf & Welling, 2016) | $\mathbf{X, A}$ | $80.9 \pm 0.4$ | $66.7 \pm 0.4$ | $77.1 \pm 0.7$ |
| VGAE (Kipf & Welling, 2016) | $\mathbf{X, A}$ | $80.0 \pm 0.2$ | $64.1 \pm 0.2$ | $76.9 \pm 0.1$ |
| DGI (Velickovic et al., 2019) | $\mathbf{X, A}$ | $81.1 \pm 0.1$ | $71.4 \pm 0.2$ | $77.0 \pm 0.2$ |
| GMI (Peng et al., 2020) | $\mathbf{X, A}$ | $82.2 \pm 0.2$ | $71.4 \pm 0.5$ | $78.5 \pm 0.1$ |
| **TopoTER** | $\mathbf{X, A}$ | $\mathbf{83.7 \pm 0.3}$ | $\mathbf{71.7 \pm 0.5}$ | $\mathbf{79.1 \pm 0.1}$ |

Table 2: Model size comparison of DGI, GMI, and the proposed TopoTER.

| Model | DGI | GMI | TopoTER |
|---|---|---|---|
| No. of Parameters | $996,354$ | $1,730,052$ | $\mathbf{736,260}$ |

# 4 EXPERIMENTS

## 4.1 NODE CLASSIFICATION

**Datasets.** We adopt three citation networks to evaluate our model: Cora, Citeseer, and Pubmed (Sen et al., 2008), where nodes correspond to documents and edges represent citations. We follow the standard train/test split in (Kipf & Welling, 2017) to conduct the experiments.

**Implementation Details.** In this task, the auto-encoder network is trained via Adam optimizer, and the learning rate is set to $10^{-4}$. We use the same early stopping strategy as DGI (Velickovic et al., 2019) on the observed training loss, with a patience of 20 epochs. We deploy one Simple Graph Convolution (SGC) layer (Wu et al., 2019) as our encoder, and the order of the adjacency matrix is set to 2, while we will study the order of the adjacency matrix in Appendix A. The LeakyReLU activation function with a negative slope of $0.1$ is employed after the SGC layer. Similar to DGI (Velickovic et al., 2019), we set the output channel $F = 512$ for Cora and Citeseer dataset, and $256$ for Pubmed dataset due to memory limitations. After the encoder, we use one linear layer to classify the transformation types. We set the edge perturbation rate in Eq. (9) as $r = \{0.7, 0.4, 0.7\}$ for Cora, Citeseer, and Pubmed, respectively. The analysis of the edge perturbation rate will be presented in Appendix B.

During the training procedure of the classifier, the SGC layer in the encoder is used to extract graph feature representations with the weights frozen. After the SGC layer, we apply one linear layer to map the features to the classification scores.

**Experimental Results.** We compare the proposed method with five unsupervised methods, including one node embedding method DeepWalk, two graph auto-encoders GAE and VGAE (Kipf & Welling, 2016), and two contrastive learning methods DGI (Velickovic et al., 2019) and GMI (Peng et al., 2020). Additionally, we report the results of Raw Features and DeepWalk+Features (Perozzi et al., 2014) under the same settings. For fair comparison, the results of all other unsupervised methods are reproduced by using the same encoder architecture of the TopoTER except DeepWalk and Raw Features. We report the mean classification accuracy (with standard deviation) on the test nodes for all methods after 50 runs of training. As reported in Tab. 1, the TopoTER outperforms all other competing unsupervised methods on three datasets. Further, the proposed unsupervised method also achieves comparable performance with semi-supervised results. This significantly closes the gap between unsupervised approaches and the semi-supervised methods.

Moreover, we compare the proposed TopoTER with two contrastive learning methods DGI and GMI in terms of the model complexity, as reported in Tab. 2. The number of parameters in our model is less than that of DGI and even less than half of that of GMI, which further shows the TopoTER model is lightweight.

Table 3: Graph classification accuracies (with standard deviation) in percentage on 6 datasets. ">1 Day" represents that the computation exceeds 24 hours. "OOM" is out of memory error.

| Dataset | MUTAG | PTC-MR | RDT-B | RDT-M5K | IMDB-B | IMDB-M |
|---|---|---|---|---|---|---|
| (No. Graphs) | 188 | 344 | 2000 | 4999 | 1000 | 1500 |
| (No. Classes) | 2 | 2 | 2 | 5 | 2 | 3 |
| **Graph Kernel Methods** | | | | | | |
| RW | $83.72 \pm 1.50$ | $57.85 \pm 1.30$ | OOM | OOM | $50.68 \pm 0.26$ | $34.65 \pm 0.19$ |
| SP | $85.22 \pm 2.43$ | $58.24 \pm 2.44$ | $64.11 \pm 0.14$ | $39.55 \pm 0.22$ | $55.60 \pm 0.22$ | $37.99 \pm 0.30$ |
| GK | $81.66 \pm 2.11$ | $57.26 \pm 1.41$ | $77.34 \pm 0.18$ | $41.01 \pm 0.17$ | $65.87 \pm 0.98$ | $43.89 \pm 0.38$ |
| WL | $80.72 \pm 3.00$ | $57.97 \pm 0.49$ | $68.82 \pm 0.41$ | $46.06 \pm 0.21$ | $72.30 \pm 3.44$ | $46.95 \pm 0.46$ |
| DGK | $87.44 \pm 2.72$ | $60.08 \pm 2.55$ | $78.04 \pm 0.39$ | $41.27 \pm 0.18$ | $66.96 \pm 0.56$ | $44.55 \pm 0.52$ |
| MLG | $87.94 \pm 1.61$ | $63.26 \pm 1.48$ | >1 Day | >1 Day | $66.55 \pm 0.25$ | $41.17 \pm 0.03$ |
| **Supervised Methods** | | | | | | |
| GCN | $85.6 \pm 5.8$ | $64.2 \pm 4.3$ | $50.0 \pm 0.0$ | $20.0 \pm 0.0$ | $74.0 \pm 3.0$ | $51.9 \pm 3.8$ |
| GraphSAGE | $85.1 \pm 7.6$ | $63.9 \pm 7.7$ | - | - | $72.3 \pm 5.3$ | $50.9 \pm 2.2$ |
| GIN-0 | $89.4 \pm 5.6$ | $64.6 \pm 7.0$ | $92.4 \pm 2.5$ | $57.5 \pm 1.5$ | $75.1 \pm 5.1$ | $52.3 \pm 2.8$ |
| GIN-$\epsilon$ | $89.0 \pm 6.0$ | $63.7 \pm 8.2$ | $92.2 \pm 2.3$ | $57.0 \pm 1.7$ | $74.3 \pm 5.1$ | $52.1 \pm 3.6$ |
| **Unsupervised Methods** | | | | | | |
| node2vec | $72.63 \pm 10.20$ | $58.58 \pm 8.00$ | - | - | - | - |
| sub2vec | $61.05 \pm 15.80$ | $59.99 \pm 6.38$ | $71.48 \pm 0.41$ | $36.68 \pm 0.42$ | $55.26 \pm 1.54$ | $36.67 \pm 0.83$ |
| graph2vec | $83.15 \pm 9.25$ | $60.17 \pm 6.86$ | $75.78 \pm 1.03$ | $47.86 \pm 0.26$ | $71.10 \pm 0.54$ | $\mathbf{50.44 \pm 0.87}$ |
| InfoGraph | $89.01 \pm 1.13$ | $61.65 \pm 1.43$ | $82.50 \pm 1.42$ | $53.46 \pm 1.03$ | $73.03 \pm 0.87$ | $49.69 \pm 0.53$ |
| **TopoTER** | $\mathbf{89.25 \pm 0.81}$ | $\mathbf{64.59 \pm 1.26}$ | $\mathbf{84.93 \pm 0.18}$ | $\mathbf{55.52 \pm 0.20}$ | $\mathbf{73.46 \pm 0.38}$ | $49.68 \pm 0.31$ |

## 4.2 GRAPH CLASSIFICATION

**Datasets.** We conduct graph classification experiments on six well-known graph benchmark datasets (Yanardag & Vishwanathan, 2015): MUTAG, PTC, REDDIT-BINARY, REDDIT-MULTI-5K, IMDB-BINARY, and IMDB-MULTI.

**Implementation Details.** In this task, the entire network is trained via Adam optimizer with a batch size of 64, and the learning rate is set to $10^{-3}$. For the encoder architecture, we follow the same encoder settings in the released code of InfoGraph (Sun et al., 2020a), *i.e.*, three Graph Isomorphism Network (GIN) layers (Xu et al., 2019b) with batch normalization. We also use one linear layer to classify the transformation types. We set the sampling rate $r = 0.5$ for all datasets.

During the evaluation stage, the entire encoder will be frozen to extract node-level feature representations, which will go through a global add pooling layer to acquire global features. We then use LIBSVM to classify these global features to classification scores. We adopt the same procedure of previous works (Sun et al., 2020a) to make a fair comparison and use 10-fold cross validation accuracy to report the classification performance, and the experiments are repeated five times.

**Experimental Results.** We take six graph kernel approaches for comparison: Random Walk (RW) (Gärtner et al., 2003), Shortest Path Kernel (SP) (Borgwardt & Kriegel, 2005), Graphlet Kernel (GK) (Shervashidze et al., 2009), Weisfeiler-Lehman Sub-tree Kernel (WL) (Shervashidze et al., 2011), Deep Graph Kernels (DGK) (Yanardag & Vishwanathan, 2015), and Multi-Scale Laplacian Kernel (MLG) (Kondor & Pan, 2016). Aside from graph kernel methods, we also compare with three unsupervised graph-level representation learning methods: node2vec (Grover & Leskovec, 2016), sub2vec (Adhikari et al., 2018), and graph2vec (Narayanan et al., 2017), and one contrastive learning method: InfoGraph (Sun et al., 2020a). The experimental results of unsupervised graph classification are preseted in Tab. 3. The proposed TopoTER outperforms all unsupervised baseline methods on the first five datasets, and achieves comparable results on the other dataset. Also, the proposed approach reaches the performance of supervised methods at times, thus validating the effectiveness of the TopoTER model.

## 5 CONCLUSION

We propose Topology Transformation Equivariant Representation (TopoTER) for learning unsupervised representations on graph data. By maximizing the mutual information between topology transformations and feature representations before and after transformations, the TopoTER enforces the encoder to learn intrinsic graph feature representations that contain sufficient information about structures under applied topology transformations. We apply the TopoTER model to node classification and graph classification tasks, and results demonstrate that the TopoTER outperforms state-of-the-art unsupervised approaches and reaches the performance of supervised methods at times.

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

## A    EXPERIMENTS ON DIFFERENT ORDERS OF THE ADJACENCY MATRIX

As presented in Sec. 3.2, we perturb the 1-hop neighborhoods via the proposed topology transformations, leading to possibly significant changes in the graph topology. This increases the difficulties of predicting the topology transformations when using one-layer GCN (Kipf & Welling, 2017) by aggregating the 1-hop neighborhood information. Therefore, we employ one Simple Graph Convolution (SGC) layer (Wu et al., 2019) with order $k$ as our encoder $E(\cdot)$, where the output feature representations aggregate multi-hop neighborhood information. Formally, the SGC layer is defined as

$$\mathbf{H} = E(\mathbf{X}, \mathbf{A}) = \left(\mathbf{D}^{-\frac{1}{2}}(\mathbf{A} + \mathbf{I})\mathbf{D}^{-\frac{1}{2}}\right)^k \mathbf{X}\mathbf{W}, \tag{16}$$

where $\mathbf{D}$ is the degree matrix of $\mathbf{A} + \mathbf{I}$, $\mathbf{W} \in \mathbb{R}^{C \times F}$ is a learnable parameter matrix, and $k$ is the order of the normalized adjacency matrix.

To study the influence of different orders of the adjacency matrix, we adopt five orders from 1 to 5 to train five models on the node classification task. Fig. 3 presents the node classification accuracy under different orders of the adjacency matrix for TopoTER and DGI respectively. As we can see, the proposed TopoTER achieves best classification performance when $k = \{4, 2, 3\}$ on the three datasets respectively. When $k = 1$, our model still achieves reasonable results although it is difficult to predict the topology transformations from 1-hop neighborhood information; when $k > 1$, our proposed TopoTER outperforms DGI by a large margin on Cora and Pubmed dataset, and achieves comparable results to DGI on Citeseer dataset. This is because DGI adopts feature shuffling to generate negative samples, which is insufficient to learn contrastive feature representations when aggregating multi-hop neighborhood information, while TopoTER takes advantage of multi-hop neighborhood information to predict the topology transformations, leading to improved performance.

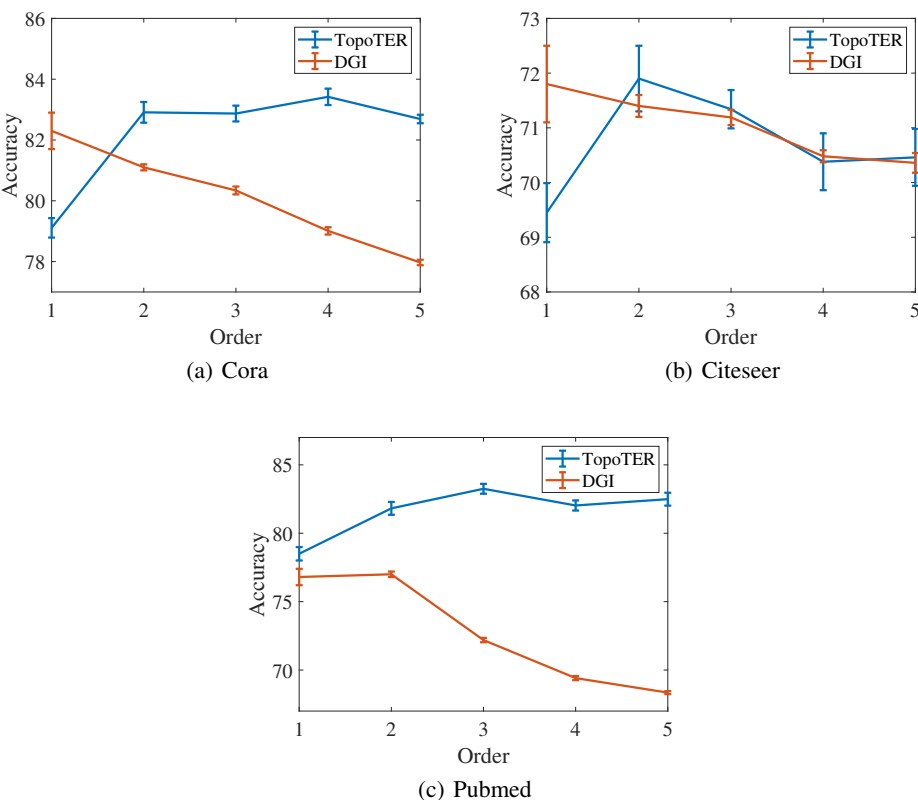

(a) Cora

(b) Citeseer

(c) Pubmed

Figure 3: Node classification accuracies under different orders of the adjacency matrix on the Cora, Citeseer, and Pubmed datasets.

# B    EXPERIMENTS ON DIFFERENT EDGE PERTURBATION RATES

Further, we evaluate the influence of the edge perturbation rate in Eq. (9) on the node classification task. We choose 11 edge perturbation rates from 0.0 to 1.0 at an interval of 0.1 to train the proposed TopoTER. We use one SGC layer as our encoder $E(\cdot)$, where the order of the adjacency matrix is set to 1. As presented in Fig. 4, the blue solid line with error bar shows the classification accuracy of our TopoTER under different edge perturbation rates. We also provide the classification accuracy on feature representations of graphs from a randomly initialized encoder $E(\cdot)$, denoted as *Random Init.*, which serves as the lower bound of the performance.

As we can see, the classification performance reaches the best when the graph is perturbed under a reasonable edge perturbation rate, *e.g.*, $r = \{0.6, 0.5, 0.6\}$ for the Cora, Citeseer, and Pubmed dataset, respectively. When the edge perturbation rate $r = 0.0$, the unsupervised training task of TopoTER becomes link prediction, which cannot take advantage of the proposed method by predicting the topology transformations; when the edge perturbation rate $r = 1.0$, our TopoTER still achieves reasonable classification results, which shows the stability of our model under high edge perturbation rates. At the same time, we observe that the proposed TopoTER outperforms *Random Init.* by a large margin, which validates the effectiveness of the proposed unsupervised training strategy.

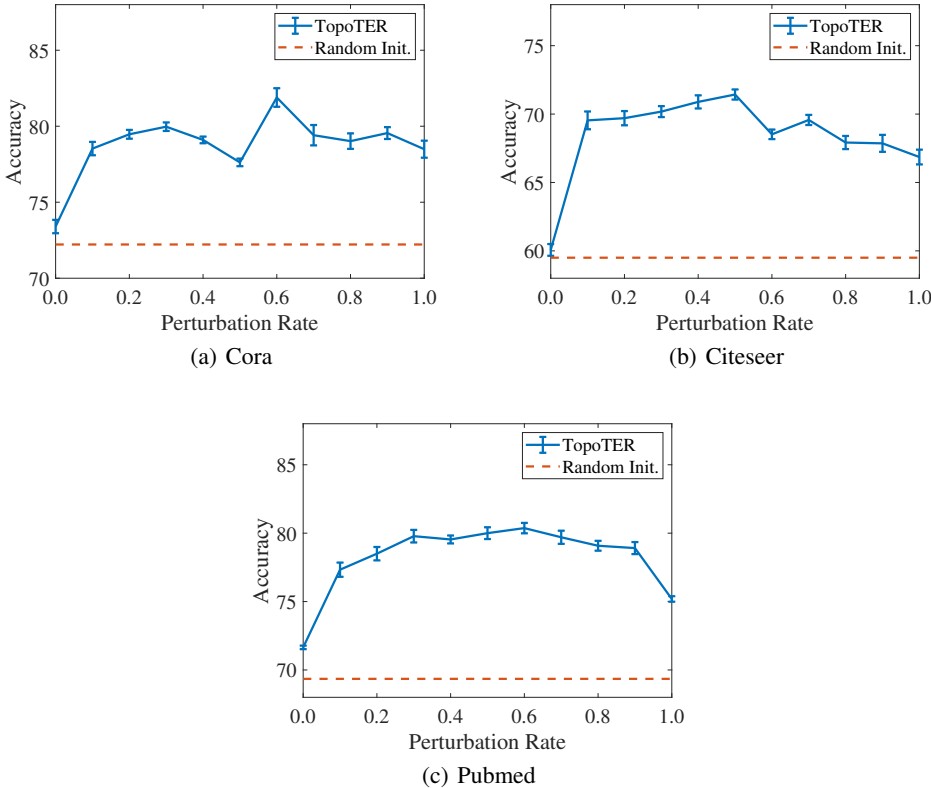

Figure 4: Node classification accuracies under different edge perturbation rates on the Cora, Citeseer, and Pubmed datasets.

