# OpenReview forum: "TopoTER: Unsupervised Learning of Topology Transformation Equivariant Representations"
_ICLR.cc/2021/Conference — Reject_

### Official Review · AnonReviewer4 · 2020-10-24
**Interesting work, similar to GraphTER**

**Rating:** 5
**Confidence:** 4

**Review:**

This paper develops a framework for unsupervised learning of graphs. The goal is to build graph representation using an encoder that is useful for downstream tasks such as graph classification. The representation is computed with an encoder $E$ applied to a graph data $(X,A)$, containing vertex data $X$ and adjacency matrix $A$. Given a graph $(X,A)$ and a perturbed version of its adjacency matrix $(X,\tilde{A)}$, the decoder $D$ is tasked with minimizing the conditional entropy, $h(\Delta A \vert H,\tilde{H})$, of the perturbation $\Delta A = A-\tilde{A}$ when given the two representations $H=E(X,A)$ and $\tilde{H}=E(X,\tilde{A})$.

This feels as a solid paper, however it is very similar to GraphTER (Gao et al. 2020), which as far as I can tell is a previous work. Let me detail the similarities. The GraphTER paper already defines the same encoder-decoder structure and training the decoder to infer the perturbation in the graph. The GraphTER  considers perturbation of node data $X$, but also perturbation of adjacency matrix since they define the adjacency matrix as a function of the node data, that is $\tilde{A}=f(\tilde{X})$. The current paper focuses only on adjacency perturbations. A second difference is that GraphTER uses the decoder to predict the perturbation directly, while the current paper asks the conditional entropy of the perturbation given the representations of the graph or its perturbation to be as small as possible (or the mutual information formulation). I don't see the benefit in the new loss formulation - if there is such a benefit it should be highlighted and demonstrated. Anyway, these two differences between the papers seem rather minor.

Other comments/questions:

-- What can be said about the equivariance of the learned encoder? Is there a way to quantify how equivariant is it? Why, if the loss that is optimized is equation 6, one would even expect equivariance? I mean, low entropy could maybe imply some unknown transformation of the perturbation. This should be explained.

-- Why not formulate the loss directly as in equation 6 (conditional entropy of the perturbation) if this is in practice what is optimized? What is the benefit in going through the mutual information formulation?

-- In the experiments part: how is the linear classifier in section 4.1 trained (after fixing the encoder)? Same question about the SVM in section 4.2? If indeed these classifiers are trained with labeled data - why is this method is called unsupervised? I am confused by that part. For example, in Table 1 it is emphasized that TopoTER does not use labels $Y$.

-- Is the encoder only one layer of GNN as in equation 11? Did you try with deeper architectures?

UPDATE: I would like to thank the authors for their rebuttal. I have read it, however unfortunately, I am not convinced the indicated differences from previous work is sufficient to warrant publication at ICLR. I am also not completely clear about the equivariance point.

---

> ### Author Response · Authors · 2020-11-19
> **We thank the reviewer for the valuable comments.**
>
> We thank the reviewer for the valuable comments.
>
> **Q: This paper is similar to GraphTER, and I don't see the benefit in the new loss formulation.**
>
> **A:** The proposed method distinguishes from GraphTER mainly in two aspects.
> 1) We formulate TopoTER from an information-theoretic perspective by maximizing the mutual information between representations and transformations, which provides a theoretical derivation for the training objective of TopoTER and generalizes transformations to more general forms.
> In contrast, GraphTER directly minimizes the MSE between the estimated and ground-truth transformations, which lacks theoretical explanation and is limited to parametric transformations.
> 2) We propose to learn how the same input signals associated with different graph topologies would lead to equivariant representations, thus enabling deeper fusion of node features and the graph topology in GCNNs. In contrast, GraphTER focuses on learning equivariant representations of nodes under node-wise transformations.
>
>
> **Q: What can be said about the equivariance of the learned encoder? Why not formulate the loss directly as in Eq. (6) (conditional entropy of the perturbation) if this is in practice what is optimized? What is the benefit in going through the mutual information formulation? If the loss that is optimized is Eq. (6), one would even expect equivariance? Is there a way to quantify how equivariant is it?**
>
> **A:** The equivariance of the learned encoder is satisfied when optimizing the loss in Eq. (6) derived from the mutual information maximization. First of all, the definition of transformation equivariance in Eq. (3), i.e., $\widetilde{\mathbf H} = \rho(\Delta\mathbf{A}) \mathbf H$, naturally leads to the maximization of the mutual information between $\widetilde{\mathbf H}$ and $(\mathbf H,\Delta\mathbf{A})$. However, it is difficult to compute the mutual information directly. We thus relax the maximization problem to cross entropy minimization in Eq. (6) for practical implementations.
> In all, the definition of transformation equivariance directly leads to the mutual information maximization, which is relaxed to Eq. (6) for efficient optimization via a rigorous proof. Thus, by optimizing the loss in Eq. (6), we would expect equivariance and quantify it through the reduction of the cross entropy (i.e., increase of mutual information) in the unsupervised training stage.
>
>
> **Q: How is the linear classifier in section 4.1 trained (after fixing the encoder)? Same question about the SVM in section 4.2.**
>
> **A:** We first adopt the unsupervisedly learned encoder with frozen weights to extract the features of the input graph. Then, we take these features and the corresponding labels as input to train the linear or SVM classifier. For the linear classifier, we use the SGD optimizer to minimize the classification loss; for the SVM classifier, we employ the Primal-Dual algorithm for optimization.
>
>
> **Q: If indeed these classifiers are trained with labeled data---why is this method is called unsupervised?**
>
> **A:** We focus on unsupervised feature learning trained without labeled data. Then, to evaluate the quality of the learned features, we apply them to downstream tasks, where SVM or linear classifiers are trained with labels for supervised evaluation following previous works.
>
>
> **Q: Is the encoder only one layer of GNN as in Eq. (11)? Did you try with deeper architectures?**
>
> **A:** We employ a one-layer GCN as our encoder in the node classification task, while we also explore deeper architectures by adopting three GNN layers as our encoder in the graph classification task for fair comparison.

---

### Official Review · AnonReviewer3 · 2020-10-28
**A method for for self-training GNN**

**Rating:** 7
**Confidence:** 3

**Review:**

The paper propose an unsupervised method for self-training of graph-neural-networks (GNNs). The authors provide information-theoretic justification to their method using maximization of the lower bound of the mutual information on their objective. Their approach is based on maximizing the mutual information between a perturbed graph topology and its node representation.

Strong points:
- very good results (some of the results are comparable to supervised method and the improvement achieved on the other unsupervised methods is significant)
- simple approach with theoretical justification
- paper is nicely written and easy to follow

---

> ### Author Response · Authors · 2020-11-19
> **We thank the reviewer for the valuable comments.**
>
> We thank the reviewer for the valuable comments.

---

### Official Review · AnonReviewer2 · 2020-10-28
**An incremental advance on unsupervised learning of node representations of graph data**

**Rating:** 6
**Confidence:** 2

**Review:**

This paper presents the Topology Transformation Equivariant Representation (TopoTER) as a learning method for the (uncommonly) unsupervised learning of node representations  in graphs, with applicability to Graph Convolutional Neural Networks.
The paper falls well within the scope of the conference.
Its writing could be improved and would benefit from a thorough language revision.
The contribution is well grounded, the state-of-the-art is appropriately identified and even though its novelty is somehow incremental, its value seems beyond doubt.
To the best of my knowledge, and not being an expert on graph data analysis, the maths seem sound.
The experimental work, including comparison with alternative approaches is quite detailed (and well-informed of the competition).
Adding extra experiments as an appendix ... I guess it verges on cheekiness, but given, the particularities of accepted formats, assume it is fine.

---

> ### Author Response · Authors · 2020-11-19
> **We thank the reviewer for the valuable comments.**
>
> We thank the reviewer for the valuable comments. We will polish the writing further and make a thorough language revision.

---

### Official Review · AnonReviewer1 · 2020-10-29
**Interesting idea & theoretical questions**

**Rating:** 6
**Confidence:** 4

**Review:**

#### Goal

- This work considers the graph task of learning node representations that are invariant to small edge perturbations. It achieves this through a data augmentation procedure that samples new “fake” edges and regularizes the GNN equivariant representations to be unable to predict these fake edges. Overall it is an interesting idea.

#### Quality

The paper is reasonably well written, except for the introduction, which throws too much jargon around without being concrete about the work’s goals.

Mathematically, I have doubts about the correctness of the work (see Cons)


#### Clarity

It is a little confusing how we can obtain the fake/deleted edge probabilities p. Earlier in the paragraph there is a connection with GAT but GAT has an attention mechanism to obtain \p_{ij}. While the first paragraph of 3.2 states the fake “edges” are given by a random matrix \Sigma obtained from a Bernoulli distribution with parameter p. I am not sure what that means. The elements of \Sigma are i.i.d. samples from a Bernoulli distribution? I am not familiar with Bernoulli distributions that sample entire random matrices.


The graph is defined with an edge set, \mathcal{E}, and an adjacency matrix A. It is unclear why we need these structures to define a single graph. Isn’t A fully determined by \mathcal{E}? If not, why not? It would be good to clarify.

- The transformation $t$ and its homomorphism \rho are given in Def 1 but not given a formal definition later in the paper. In the proposed algorithm in Section 3.4, what are t and \rho?

#### Originality

There is some level of originality, although there are too many papers in GNNs doing similar things to know for sure. It claims to be an edge extension of a previous work for nodes.


#### Significance

Overall, the work does not really bring much to the table. Robustness to a certain type of edge noise could be interesting, but the choice of noise process is probably very important. The noise process is not discussed in depth. Data augmentation (just adding the new perturbed graph as a data example) is the main competitor to this approach. The difference between what the authors did and data augmentation should be front and center at the experimental evaluation.


#### Pros

- The paper central idea applies invariant representations to make the GNN representation more robust to noise

#### Cons

- Predicting edges with equivariant node representations is provably impossible (Srinivasan & Ribeiro, 2020). The Representation Encoder looks like an equivariant node representation to me. Hence, the “Transformation Decoder” is attempting a provably impossible task. Remark 1 in (Srinivasan & Ribeiro, 2020) gives an interesting comment to why GraphSAGE is successful even if it was trying the same impossible task. The authors need to give a solid theoretical reason for their approach, otherwise it is not believable.

The main competitor of this approach is data augmentation. GraphSAGE has an unsupervised version that also adds fake edges at random as a form of data augmentation.

Srinivasan, Balasubramaniam, and Bruno Ribeiro. "On the Equivalence between Positional Node Embeddings and Structural Graph Representations." In International Conference on Learning Representations. 2020.


#### Other comments

- In Franceschi et al., 2019,  the authors also want to learn the underlying graph (which edges are fake, which are true). It seems that their approach could be applied in the framework of this work, and would maybe give a better way to sample the fake edges.

Franceschi, Luca, Mathias Niepert, Massimiliano Pontil, and Xiao He. "Learning discrete structures for graph neural networks." ICML (2019).

---

> ### Author Response · Authors · 2020-11-19
> **We thank the reviewer for the valuable comments and references.**
>
> We thank the reviewer for the valuable comments.
>
> **Q: It is a little confusing how we can obtain the fake/deleted edge probabilities $p$. The elements of $\Sigma$ are i.i.d. samples from a Bernoulli distribution?**
>
> **A:** We sample the elements $\sigma_{i,j}$ of $\Sigma$ from a Bernoulli distribution $\mathcal{B}(p)$. $\sigma_{i,j}$ is a random variable which takes the value 1 with probability $p$ and the value 0 with probability $1-p$, where the probability $p$ is a hyperparameter specified by ourselves for different edge perturbation rates.
>
> **Q: It is unclear why we need both the edge set and adjacency matrix to define a single graph. Isn't $\mathbf{A}$ fully determined by $\mathcal{E}$?**
>
> **A:** In our paper, the edge set $\mathcal{E}$ contains all connected node pairs, while the adjacency matrix $\mathbf{A} \in R^{N \times N}$ is a binary matrix to indicate whether two nodes are connected. The edge set and adjacency matrix can be determined by each other for unweighted graphs. However, it is general to define both $\mathcal{E}$ and $\mathbf{A}$ especially for weighted graphs.
>
> **Q: In the proposed algorithm in Section 3.4, what are $\mathbf{t}$ and $\rho(\mathbf{t})$?**
>
> **A:** In Section 3.4, the transformation $\mathbf{t}$ applied to $\mathbf{A}$ is $\mathbf{A}+\Delta\mathbf{A}$, as defined in Eq. (2) before Def 1. Meanwhile, $\rho(\mathbf{t})$ reflects the equivariance in the feature space as the graph transforms by the transformation $\mathbf{t}$. It is implicitly contained in the feature representations before and after transformation, and thus we can decode the transformation $\mathbf{t}$ from the representations to ensure equivariance. We will make this definition clear.
>
> **Q: It claims to be an edge extension of a previous work for nodes.**
>
> **A:** The proposed method distinguishes from a previous work GraphTER mainly in two aspects.
> 1) We formulate TopoTER from an information-theoretic perspective by maximizing the mutual information between representations and transformations, which provides a theoretical derivation for the training objective of TopoTER and generalizes transformations to more general forms.
> In contrast, GraphTER directly minimizes the MSE between the estimated and ground-truth transformations, which lacks theoretical explanation and is limited to parametric transformations.
> 2) We propose to learn how the same input signals associated with different graph topologies would lead to equivariant representations, thus enabling deeper fusion of node features and the graph topology in GCNNs. In contrast, GraphTER focuses on learning equivariant representations of nodes under node-wise transformations.
>
> **Q: Robustness to a certain type of edge noise could be interesting, but the choice of noise process is probably very important, which is not discussed in depth.**
>
> **A:** Thanks for your valuable comments. In our paper, we propose to learn equivariant representations of graphs under topology transformation by adding or removing edges from graphs. Other choices of topology transformations include adding noise to edge weights in a weighted graph or filtering the entire graph. However, we focus on unweighted graphs in this paper, while weighted graphs will be explored as our future work.
>
> **Q: Predicting edges with equivariant node representations is provably impossible. The authors need to give a solid theoretical reason for their approach, otherwise it is not believable.**
>
> **A:** Actually, instead of predicting edges from node representations, we predict the "transformations in edges" from the "difference" between feature representations of graph signals before and after transformation. As presented in Eq. (10), (11), and (12) of our manuscript, the difference between $\widetilde{\mathbf{H}}$ and $\mathbf{H}$ captures the information of the topology transformation $\Delta\mathbf{A}$, which enables us to predict the topology transformation from the difference in node representations before and after transformation.
>
> **Q: GraphSAGE has an unsupervised version that also adds fake edges at random as a form of data augmentation. The difference between what the authors did and data augmentation should be front and center at the experimental evaluation.**
>
> **A:** Thank you for the comments. However, this paper is not aimed at invariant representations that make GNNs more robust to noise, but exploits representations that are EQUIVARIANT under topology transformations, e.g., adding or removing edges. Hence, fake edges in our paper are designed for the learning of equivariant representations by predicting topology transformations (including fake edges) in an unsupervised fashion.
>
> **Q: In Franceschi et al., 2019, the authors also want to learn the underlying graph (which edges are fake, which are true).**
>
> **A:** Thanks for this comment. It would be beneficial to use their method for sampling fake edges.

---

### Public Comment · ~Yuning_You1 · 2020-11-10
**Interesting perspective and related work**

We would like to draw your attention that we have related work demonstrates the power of self-supervision and contrastive learning with augmentations in the graph domain. We hope to have further discussions and references with you.

When Does Self-Supervision Help Graph Convolutional Networks? ICML 2020. https://arxiv.org/abs/2006.09136

Graph Contrastive Learning with Augmentations, NeurIPS 2020. https://arxiv.org/abs/2010.13902

---

> ### Author Response · Authors · 2020-11-19
> **Many thanks for your interest in our work**
>
> Thank you for your interest in our work and for providing related works of self-supervised learning in the graph domain.
> The related two papers study self-supervised graph representation learning from other interesting perspectives such as contrastive learning and demonstrate the power of self-supervision in the graph domain, which are good additions to our related work. It would be great to have further discussions with you!

---

### Decision · Program_Chairs · 2021-01-07
**Final Decision**

**Decision:**

Reject

**Comment:**

The paper is concerned with learning transformation equivariant node representation of graph data in an unsupervised setting. The paper extends prior work in this topic by focusing on equivariance under topology transformations (adding/removing edges) and considering an information theoretic perspective. Reviewers highlighted the promising ideas of the approach, its relevance for the ICLR community, and the promising experimental results (although improvements over prior work are not necessarily significant on all benchmarks).

However, reviewers raised concerns regarding the novelty of the method and the clarity of presentation with respect to key parts of the method. These aspects connect also to further concerns raised, e.g., related to mathematical correctness as well as the significance of the proposed loss function, the benefits of motivating it from MI, and the improvements over GraphTER. The rebuttal didn't fully clarify these points. While the paper is mostly solid, I agree with the reviewers' concerns and -- currently -- the paper doesn't clear the bar for acceptance; it would require another revision to improve upon these points. However, I'd encourage the authors to revise and resubmit their work with considering this feedback.